# Effect of Substitution Degree and Homogeneity on Cyclodextrin-Ligand Complex Stability: Comparison of Fenbufen and Fenoprofen Using CD and NMR Spectroscopy

**DOI:** 10.3390/ijms24087544

**Published:** 2023-04-19

**Authors:** Márta Kraszni, Ferenc Ágh, Dániel Horváth, Arash Mirzahosseini, Péter Horváth

**Affiliations:** 1Department of Pharmaceutical Chemistry, Semmelweis University, Hőgyes Endre utca 9, 1092 Budapest, Hungary; kraszni.marta@semmelweis.hu (M.K.); mirzahosseini.arash@semmelweis.hu (A.M.); 2ELKH-ELTE Protein Modelling Research Group, Eötvös Loránd University, Pázmány Péter sétány 1A, 1117 Budapest, Hungary; daniel.horvath@ttk.elte.hu

**Keywords:** cyclodextrin, host–guest complex stability, induced circular dichroism, NMR, fenbufen, fenoprofen

## Abstract

The stability of host–guest complexes of two NSAID drugs with similar physicochemical properties, fenbufen and fenoprofen, was investigated by comparing induced circular dichroism and ^1^H nuclear magnetic resonance methods using eight cyclodextrins of different degrees of substitution and isomeric purity as guest compounds. These cyclodextrins include native β-cyclodextrin (BCyD), 2,6-dimethyl-β-cyclodextrin 50 (DIMEB50), 80 (DIMEB80) and 95% (DIMEB95) isomerically pure versions, low-methylated CRYSMEB, randomly methylated β-cyclodextrin (RAMEB) and 4.5 and 6.3 average substitution grade hydroxypropyl-β-cyclodextrin (HPBCyD). The stability constants obtained by the two methods show good agreement in most cases. For fenbufen complexes, there is a clear trend that the stability constant increases with the degree of substitution while isomer purity has a smaller effect on the magnitude of stability constants. A significant difference was found in the case of DIMEB50 when compared to DIMEB80/DIMEB95, while the latter two are similar. In the fenbufen–fenoprofen comparison, fenbufen, with its linear axis, gives a more stable complex, while fenoprofen shows lower constants and poorly defined trends.

## 1. Introduction

Cyclodextrins (CyDs) are used in many areas of the pharmaceutical and food industry [1,2,3,4,5,6], some of their most important properties being that they can enhance the dissolution of poorly water-soluble substances, reduce the volatility of volatile molecules, and increase the chemical stability of degradable substances by inclusion complexation. As an indication of the importance of CyDs, the number of publications on CyDs in the Scopus database is increasing year by year. In pharmaceutical applications, they can also influence the pharmacokinetics of active substances and can be used as processing excipients and orphan drugs [7]. Recently, a review of NSAID active substances was published [8] in which the literature of about 60 different inclusion complexes of 24 active substances and 12 types of CyDs was reviewed between 2010 and 2020. CyDs are also often used as chiral selectors in the analysis of chiral molecules (enantiomers) in separation processes [9]. Chiral differentiation and structure elucidation of cyclodextrins have been extensively studied in the literature [10,11,12,13,14,15,16]. In addition to the natural (α, β, γ) CyDs, a number of substituted derivatives are known that affect the properties of the original molecule.

The substituents may be neutral (methyl, hydroxypropyl), positively charged (e.g., amino-, guanidino- [16]), or negatively charged (carboxymethyl, sulfobutyl-ether, sulfated [17]) derivatives, depending on the intended purpose. In the case of substituted derivatives with acid–base active functional groups, the average protonation state is pH-dependent and the determination of the p*K*_a_ values for each substituent is not straightforward [18]. Substitution can be made on the OH groups 2, 3, and 6 of α-D-glucopyranose units, resulting in mono-, di-, or trisubstituted derivatives. The wider rim of the CyD ring contains the secondary OH groups at position 2 and 3, while the narrower rim of the truncated cone-shaped ring system contains the primary OH groups at position 6. Thus, a β-cyclodextrin (BCyD) with 1–4 glycosidic OH bonds from seven sugar subunits can theoretically have 7 substituents on the molecule in a monosubstituted, 14 in a disubstituted, and 21 in a trisubstituted state. The selectivity of the substitution reaction strongly influences the actual substitution site and the number of substituents [19,20]. It is therefore important that the average number of substituents (DS—Degree of Substitution) and the isomeric purity of the resulting product are determined by appropriate analytical methods [21]. In the period of 2018–2022, Scopus returned only 136 publications for “single isomer” and “cyclodextrin” out of nearly 100,000 hits related to cyclodextrins.

Several methods are available for the qualitative and quantitative characterization of CyD complexes in both solution and solid phases. The most important features are the stoichiometry of the complex (C), i.e., the host (H) to guest (G) ratio, and the stability constant (log*K*). In our previous work, the stability constants of CyD complexes of antifungal azoles were determined by circular dichroism (CD) spectroscopy and the results were compared with literature data [22] in Table 1. A relatively good agreement was obtained for the native BCyD, but quite significant differences were found for the substituted derivatives.

Differences between stability constants determined by different methods may be due to: (1) in the comparison, the degree of substitution and isomeric composition of the substituted CyD used are not precisely defined; (2) in solution phase experiments, different experimental conditions may cause discrepancies, e.g., temperature, pH, ionic strength, or the presence of other components.

Recognizing these differences, we set out to determine the stability constants of CyD derivatives with different degrees of substitution and isomeric purity for two NSAIDs (fenbufen and fenoprofen depicted in Figure 1) with similar physicochemical properties by CD and ^1^H NMR spectroscopy. The model compounds were selected according to the following criteria. On the one hand, most NSAIDs are poorly soluble in water. This can lead to problems with bioavailability. An important role of cyclodextrins is to improve the solubility of poorly soluble compounds. Several formulations containing NSAID-cyclodextrin complexes are currently on the market, e.g., Meloxicam/Mobitil; Nimesulide/Nimedex; Piroxicam/Cycladol or Brexin; Diclofenac/Voltaren Ophtalmic, and in this way, the study may have practical relevance in the future. On the other hand, the ICD signal requires the guest molecule to have a chromophore group, a condition that is met by both compounds. The main consideration in the design of the experiments was to measure the same solutions with the two orthogonal methods. Bratu et al., investigated the BCyD-fenbufen system by NMR, and determined the stoichiometry, structure, and stability constant. They reported a 1:1 composition and a value of “several 1000 s (M^−1^)” [30]. Another literature value could be found (4.72 × 10^3^) in the publication of Zhang et al. [31] where mass spectrometry was applied to determine the stability of the complex. Furthermore, UV-, induced CD-, and fluorescence spectroscopic methods were used by Sortino et al. [32] to determine the BCyD-fenbufen stability constant. They determined *K* values of 3200 M^−1^ by UV, 1690 and 2200 M^−1^ at two different wavelengths by induced CD and 2690 M^−1^ by fluorescence. The stability of a pure *S*-isomer of the BCyD-fenoprofen complex was determined by ^1^H NMR spectroscopy [33]. Uccello-Barretta et al., found a 1:1 stoichiometry and a *K* value of 2010 M^−1^. They calculated the result from the chemical shift of the methyl proton.

^1^H NMR spectroscopy can be used to monitor complexation as the signals of the guest and CyD are usually visible. The NMR spectra of the investigated guest molecules and BCyD together with their assignments are shown in Figure 2.

The change in the chemical shift of the guest signals as complexation occurs (CIS—Complexation Induced Shift) can be used to determine the stability constant.
*δ*_obs_ = *δ*_G_∙χ_G_ + *δ*_C_∙χ_C_(1)

As shown in Equation (1), the observed chemical shift (*δ*_obs_) is the average of the chemical shift of the free guest (*δ*_G_) and the complexed (*δ*_C_) species weighted by their mole fraction (*χ*). In addition to the stability constants, the stoichiometry of the complex can also be estimated [34,35] and information on the structure of the complex can be obtained from the NOESY or ROESY spectra, if cross peaks are observed between the guest signals and the CyD H3 and H5 are found inside the cavity. The limitation of NMR is a low signal-to-noise ratio and an inability to discern between different CyD isomers. In case of native cyclodextrins—due to their symmetry—the interpretation of the NMR spectra can be simple, but in the case of modified cyclodextrins, the presence of structural isomers (different substitution degree and position) complicates the structural analysis. Although CD spectroscopy yields less information in terms of exact atomic interactions, it has superior selectivity towards the complex formation, and, as such, is one of the most effective methods for CyD complex analysis [36]. The basis of the CD selectivity is the fact that achiral guests have no CD spectra while CD spectra of CyD is insignificant above 220 nm. Therefore, the requirement for an induced CD spectrum (ICD), which solely originates from the complex, is a chromophore (of the guest) that is under chiral perturbation by the CyD [37]. The appearance of this ICD signal in the region of the absorption bands is a definite proof of the complex formation. A further requirement is that the dipole moment of the electron transition of the chromophore and the axis of the cyclodextrin have a favorable geometric alignment since the rotator strength is derived from these two factors. It may be that the geometry of the complex is so unfavorable that no ICD signal appears. The sign and intensity of the ICD signal can provide information regarding the structure of the complex [38,39,40]. The value of the stability constant can be obtained from logistic non-linear regression on the ICD ~ host concentration data (with constant guest concentration), as described in our previous study [22]. A limitation of CD spectroscopy is the requirement for the presence of a chromophore on the guest and the low intensity of ICD signals when absorption is relatively high. This latter effect is a result of the absorption difference (Δ*ε*) of the two circularly polarized light components being proportional to the CD signal, while the sum of the component absorptions is equal to the total absorption. Another disadvantage is that the intensity of the ICD signal is usually an order of magnitude lower than the normal CD signal, and for racemic compounds, only an apparent stability constant can be calculated for the two enantiomers. The intensity of the ICD signal can be influenced by the dynamics of the complex and by possible hydrogen bond interactions.

The use of cyclodextrins in pharmaceutical research and development is becoming increasingly frequent. Accurate characterization of cyclodextrin complex stability constants is essential for correct drug formulations. Of the analytical methods commonly used to characterize cyclodextrin complexation, two well-known methods are CD and NMR, but in some cases (see Table 1), a considerable difference has been observed between the stability constants obtained by the two methods. The purpose of this study was to investigate whether this difference is relevant and systematic. We chose to perform this investigation on two selected guest compounds and a variety of CyD hosts.

## 2. Results

### 2.1. Fenbufen Complex Stability Constants

Fenbufen complexation with the various CyD hosts was analyzed using CD and ^1^H NMR spectroscopy. The ICD spectra of HPBCyD(4.5)-fenbufen are shown in Figure 3, along with the ^1^H NMR spectra series. Figure 3 also depicts the non-linear regression plots on the CD and NMR data. The CD and NMR data with the other CyDs can be found in Appendix A. The determined complex stability constant values in log10 units are summarized in Table 2, together with the difference between the values obtained with the two methods. Furthermore, in Table 3, the non-linear regression results from each individual NMR signals are compiled, where the log*K** is handled as a random effects model parameter as opposed to the log*K* fitted globally to all NMR signals as a fixed effects model parameter presented in Table 2.

### 2.2. Fenoprofen Complex Stability Constants

Fenoprofen complexation was studied with the same methods as for fenbufen. The ICD and ^1^H NMR spectra of CyD-fenoprofen complexes are shown in Figure 4 along with select non-linear regression plots. The full set of CD and NMR data are found in Appendix A. The determined complex stability constant values (and the difference between the methods) in log10 units are compiled in Table 4. In Table 5, the non-linear regression results from each individual NMR signals are tabulated.

The BCyD-fenoprofen complex was also studied with ROESY NMR in order to elucidate the binding points in the complex. Figure 5 shows the ROESY cross peaks between the H3, H5, and H6 hydrogens of BCyD and the aromatic hydrogens of fenoprofen.

## 3. Discussion

The combined CD and NMR analysis of CyD-fenbufen and CyD-fenoprofen complexes revealed that in the case of both complexes, the aromatic rings of the guest species entered the hydrophobic cavity of the cyclodextrin hosts. Fenbufen showed similar ICD spectra for all CyD hosts, as seen in Figure 3, with a characteristic spectra series of HPBCyD(4.5) complex. For the fenbufen-cyclodextrin complex, a relatively high intensity ICD signal with a positive sign was obtained for all cyclodextrins. Thus, based on the Kodaka–Harata rules [38,40] it can be assumed that the temporary dipole vector of the excitation transition and the axis of the cyclodextrin cavity coincide, and the chromophore of the guest molecule is predominantly located in the cyclodextrin cavity.

On the other hand, fenoprofen had different ICD profiles for the various cyclodextrins (Figure 4). Only the BCyD complex had higher intensity positive bands, whereas CRYSMEB had only about half the intensity. These ICD bands still suggest that the chromophore is predominantly located in the cyclodextrin cavity, but in the case of CRYSMEB the methyl groups at the rim already inhibit the complete immersion. For the other cyclodextrins, the lower intensity bands make it more difficult to predict the structure. The substituents on the cyclodextrin rim may be spatially inhibited, resulting in less immersion. In the absence of quantum chemical calculations, the temporary dipole vector of the excitation transition is also difficult to estimate but is likely to be non-parallel to the axis of the cyclodextrin cavity. It is also difficult to describe only one structure based on 2D NMR (ROESY) spectra, since the H2, H3, and H6 hydrogens of BCyD gave rise to several cross-peaks with the aromatic protons of fenoprofen (Figure 5). NMR and CD results suggest that the ICD signal is lower in intensity due to the complex dynamics and that several structures may exist simultaneously. The spectral differences between fenbufen- and fenoprofen-CyD complexes are probably due to fenbufen having a rather linear shape and thus not having many degrees of freedom when the biphenyl part of the molecule enters the CyD cavity, while fenoprofen can enter cyclodextrins in various ways as a result of the rotating etheric oxygen bridge.

When comparing the complex stability constants obtained from NMR measurements, one can perform a non-linear regression analysis on the data of the NMR signals with the log*K* as a random effects model parameter, i.e., allowing different log*K* values for the regression fit of individual NMR peak data (Table 3 and Table 5). This sometimes leads to considerable differences (or even the fit did not converge for a given NMR signal) in the obtained log*K* values for the same complex. This can be interpreted mainly as an error effect of certain NMR signals since the deviance of log*K* values obtained from individual NMR peaks from a globally fitted log*K* showed a direct relationship with the response effect (maximal chemical shift response) of that individual NMR peak (Figure 6). That is, the greater the response of an NMR signal to complexation, the greater the reliability of log*K* obtained from that signal. It could also be argued that different NMR signals are sensitive to different aspects of the complexation; therefore they might convey different information and thus different apparent log*K* values are obtained. It is therefore much more reliable to perform the regression analysis of the NMR data globally, i.e., holding the log*K* as a fixed effects model parameter only and as a result the same log*K* value is obtained from the fit of all NMR signals. This globally fitted log*K* will have a lower standard error and a better agreement with the regression result of CD data. It may be argued therefore, that CD measurements represent the complexation, as a whole since the chromophore of the guest is central to the binding processes (H-bond formation, hydrophobic interactions) and is therefore perturbed by all aspects of complexation.

The complex stability constants obtained from the two methods (CD and NMR global fit) can be compared with a Bland–Altman plot (Figure 6), where the mean of log*K*_CD_ and log*K*_NMR_ values obtained for a particular complex are plotted against the differences (log*K*_CD_ − log*K*_NMR_). From this scatter plot one can evaluate the deviation of mean differences (i.e., bias) from the null value and also the 95% confidence interval of the differences gives us an idea on whether the methods differ considerably in variability inherent to the measurement.

It can be seen from the Bland–Altman plot that the 95% confidence interval of the differences falls within 0.2 log*K* units; given the uncertainty of the obtained data from CD measurements and global fits, it can be stated that this difference between the two methods is not considerable, i.e., methodologically not relevant. There is also no tendency in the scatter of the points in the Bland–Altman plot and no considerable systematic error (bias). The small differences between the two methods can thus be explained with measurement error (differences in the sensitivity of the two methods). The larger difference found in the literature review (Table 1) can therefore be assigned to varying experimental procedures or conditions, such as solvent composition, concentrations achieved during the measurements, or poorly defined substitution degree of CyD.

It is also revealed by the data in Table 2 and Table 4 that the complex stability constants for both guests show a tendency towards lower values for BCyD. In the case of fenbufen, among the methyl substituted CyD hosts, the stability increases with the substitution grade, and also with homogeneity. Among the HPBCyD hosts, the stability increases with the substitution degree. The trends are much less pronounced for fenoprofen.

## 4. Materials and Methods

### 4.1. Materials

All cyclodextrins used were products of Cyclolab Ltd. (Budapest, Hungary). Cyclolab Ltd. have provided us with quality certificates for the batches in question: BCyD—beta-cyclodextrin; CRYSMEB randomly methylated methyl-beta-cyclodextrin with low average substitution degree (/CYL4429/DS = 4.9; Purity > 99%); DIMEB—2,6-Di-O-methyl-beta-cyclodextrin as 3 differently derivatized substance: (1). DIMEB50 (/CYL4477/DS = 15.8, Purity > 95%, Isomer purity 35,7%); (2). DIMEB80 (/CYL2305/DS = 14, Purity > 95%, Isomer purity 80%); (3). DIMEB95 (/CYL4622/DS = 14.5, Purity > 99%, Isomer purity 93,4%); RAMEB—Random methyl-beta-cyclodextrin (/CYL4537/DS = 11.2, Purity > 99%); HPBCYD (2-Hydroxypropyl)-beta-cyclodextrin: (1). HPBCyD(4.5) (/CY-2005/.2.27 DS = 4.5); (2). HPBCyD(6.3) (/CY-2005/DS = 6.3).

Fenbufen (γ-Oxo-(1,1′-biphenyl)-4-butanoic acid) and fenoprofen calcium salt (calcium (±)-2-(3-phenoxyphenyl)propanoate dihydrate (Figure 2) as well as methanol-*d*_4_ (CD_3_OD), D_2_O and sodium-deuteroxide (NaOD) were purchased from Sigma-Aldrich/Merck Group (Budapest, Hungary).

### 4.2. Preparation of Solutions

To ensure that the conditions were identical, both the CD and NMR measurements were carried out on the same solutions. The solutions had to be prepared so that the concentration was high enough for the less sensitive NMR measurements, but not so high to produce absorbance values of higher than 2 during the CD measurements in order to avoid increasing spectral noise. As a first step, stock solutions of the two NSAIDs and of the CyDs were prepared.

Fenbufen stock solution of 10 mM was prepared in D_2_O by adding a small amount of NaOD solution to aid dissolution. Since fenoprofen calcium salt cannot be dissolved in pure water, methanol-*d*_4_ was used to prepare a 4 mM stock solution. Cyclodextrin stock solutions were prepared in H_2_O (22–25 mM for fenbufen and 12 mM for fenoprofen).

The difference in concentration between the solutions of the two guest molecules is due to the differing chromophores (Figure 7). Whereas in fenbufen, the conjugation of the biphenyl group and the carbonyl oxygen gives a broad, high intensity absorption band at 225–325 nm, in fenoprofen, the two aromatic rings are separated by an etheric oxygen. This results in a characteristic triple band and low intensity spectrum for the aromatic L_b_ (benzoid) band.

For the measurements, a series of solutions was prepared, each containing the same amount of fenbufen or fenoprofen, but different volumes of CyD stock solution. The final volume of each solution was 1.00–1.50 mL for fenbufen and 5.00 mL for fenoprofen. The ratio of guest molecule to CyD varied between 1:0.25 and 1:20 and all the solutions contained approximately 10% deuterated solvent (D_2_O in fenbufen solutions and methanol-*d*_4_ in fenoprofen solutions) which provided the lock signal in NMR measurements. As methanol does not have significant interaction with CyDs [41,42] small amount of methanol was also added to the fenbufen solution as an NMR chemical shift reference standard. In the measured samples, the concentration of fenbufen was about 1 mM and that of fenoprofen 0.4 mM. For all cases, the spectrum of an aqueous solution containing the same concentration of NSAID without CyD was also recorded. The exact concentrations of the CyDs and guest compounds in the solution series can be found in Appendix A.

### 4.3. Circular Dichroism (CD) Measurements

CD and UV experiments were performed on a Jasco J-815 spectrometer (Jasco LTD, Tokyo, Japan) in cylindrical Hellma cuvettes with pathlengths of 0.1 cm for fenbufen and 1.0 cm for fenoprofen. The slit was set to 1 nm, the registration speed was set to 50 nm/min, and the accumulation was set to 5. In order to obtain accurate readings of the measured ICD values, noise filtering was performed using the Fast Fourier Transform menu in Spectra Analysis program. For fenbufen, the ellipticity data were read at the peak maximum, for fenoprofen, at the 2 or 3 peak maxima (the number of maxima depends on the type of the CyD).

### 4.4. Nuclear Magnetic Resonance (NMR) Measurements

^1^H NMR spectra were recorded at 25 °C on a Varian (Varian, Palo Alto, California, United States) DDR spectrometer (599.9 MHz for ^1^H) equipped with a dual 5 mm inverse detection gradient (IDPFG) probe-head with z-gradient. ^1^H NMR spectra were collected in 32,000 data points using 32 scans with a spectral window of 6000 Hz and referenced to the methanol (fenbufen solutions) or the methanol-*d*_4_ (fenoprofen solutions) signals. For the suppression of the water signal, the double pulse field gradient spin echo (DPFGSE) sequence was used.

The 2D ROESY spectrum of the fenoprofen-BCyD 1:5 complex was recorded at 25 °C using Bruker (Bruker, Billerica, Massachusetts, United States) Avance III spectrometer (700.1 MHz for 1H) equipped with a 5 mm inverse TCI probehead with z-gradient. The Bruker rosygpph19 pulse sequence was used with a mixing time of 300 ms and a spectral width of 10 ppm in both dimensions. The spectrum was acquired into 1024 complex points in t2 with 128 scans coadded at each of 256 t1 increments. The spectrum was processed to 2048 × 1024 data points.

### 4.5. Mathematical Equations

For the regression analysis, the following relationships were taken into account. The equilibrium constant (stability constant, *K*) that describes the formation of the complex (C) from 1:1 stoichiometric association of the host (H) and guest (G) can be written as follows:(2)$$K=\frac{\left[C\right]}{\left[H\right]\cdot \left[G\right]}$$

The equilibrium concentrations [ ] in the above equation can be expressed in terms of total concentrations [ ]_T_ as follows:(3)$$\left[H\right]={\left[H\right]}_{T}-\left[C\right]$$
(4)$$\left[G\right]={\left[G\right]}_{T}-\left[C\right]$$

Substituting Equations (3) and (4) into Equation (2) and rearranging the equation one affords a quadratic solution for the complex concentration:(5)$$\left[C\right]=\frac{{\left[G\right]}_{T}+{\left[H\right]}_{T}+\frac{1}{K}-\sqrt{{\left({\left[G\right]}_{T}+{\left[H\right]}_{T}+\frac{1}{K}\right)}^{2}-4\cdot {\left[G\right]}_{T}\cdot {\left[H\right]}_{T}}}{2}$$

In CD measurements, the ellipticity depends only on the amount of complex. Since the concentration of guest is kept constant during the experiments, at excess host concentrations, the concentration of the complex is essentially equal to the total guest concentration; as a result, the maximal ellipticity response (*θ*_max_) can be written as:(6)$$\theta_{max}\propto {\left[G\right]}_{T}$$

For ellipticity values below the maximum Equation (7) holds:(7)$$\theta =\theta_{max}\cdot \frac{\left[C\right]}{{\left[G\right]}_{T}}$$

For the regression analysis of NMR data the following analogous Equation (8) was applied, that was obtained from Equation (1), where Δ*δ* is the Complexation Induced Shift (i.e., *δ*_obs_ − *δ*_G_), and Δ*δ*_max_ is the maximal chemical shift response (i.e., *δ*_C_ − *δ*_G_).
(8)$$∆\delta ={∆\delta }_{max}\cdot \frac{\left[C\right]}{{\left[G\right]}_{T}}$$

The non-linear regression analyses were performed using Origin Pro 8 (OriginLab Corp., Northampton, MA, USA). The standard errors of the fitted parameters were used to calculate the Gaussian propagation of uncertainty to the other parameters derived in the Results chapter. The Bland–Altman plot was constructed based on guidelines of the authors [43] in R version 4.1.3 (R Foundation for Statistical Computing, Vienna, Austria) [44].

## 5. Conclusions

The main purpose of this study was to investigate two methods used for characterizing cyclodextrin complexes: CD and NMR spectroscopy. The difference between the two methods in the determination of CyD-complex stability constants is not significant, as expected. The ICD data provide selective information on the complex formation and the concomitant binding perturbations. For ICD spectra with one maximum, it is advised to obtain the stability constant from the ellipticity data at the maximum. However, for structured ICD spectra, as is the case for fenoprofen, ellipticity data readings at multiple wavelength maxima are more appropriate followed by a global fit with mixed effects. Regression analysis of NMR data requires more precaution; the appropriate signals must be chosen based on the CIS magnitude. The substitution degree of CyD and their isomeric purity influences the stability constant. For both guests, native BCyD affords one of the lowest stability constants, while the substitution degree and isomeric purity increase the stability constant. For fenbufen, the order of stability constants is as follows:

BCyD ≈ CRYSMEB < HPBCyD4.5 < RAMEB ≈ HPBCyD6.3 ≈ DIMEB50 < DIMEB80 ≈ DIMEB95.

No similar trend was found for fenoprofen, although slightly higher constants were obtained for CyD derivatives with higher degrees of substitution. In the comparison between the guests (fenoprofen–fenbufen), it is clear that due to its linear shape, fenbufen forms more stable complexes compared to fenoprofen, since in the latter, the two aromatic rings are connected via an etheric oxygen, granting the molecular structure higher degree of freedom.

The results of the present study show that stability constants should only be compared if the average degree of substitution, the isomeric purity, and the measurement conditions are well defined. Our measured stability constant for the fenbufen-BCyD complex is in excellent agreement with a literature value measured by ICD by Sortino et al. [32] and in good agreement with constants measured by other methods [31,32]. For the fenoprofen-BCyD complex, our measured stability constant also shows good agreement with the literature value determined by ^1^H NMR [33].

## Figures and Tables

**Figure 1 ijms-24-07544-f001:**
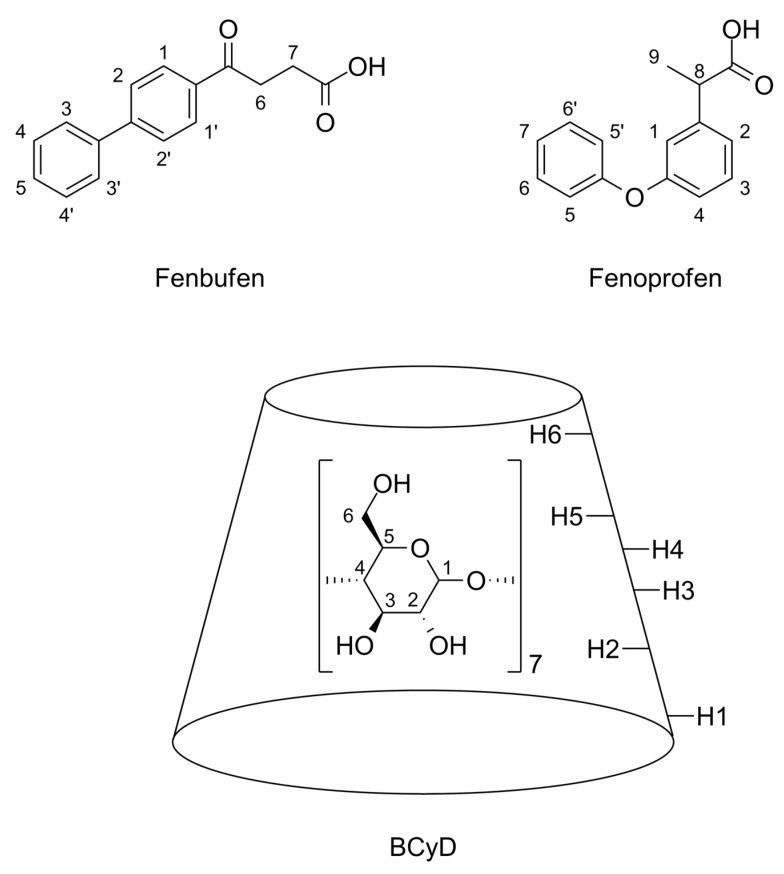
The structural formulae of the studied guest compounds (fenbufen and fenoprofen racemate) and the BCyD host compound, together with the numbering of H atoms of interest for ^1^H NMR.

**Figure 2 ijms-24-07544-f002:**
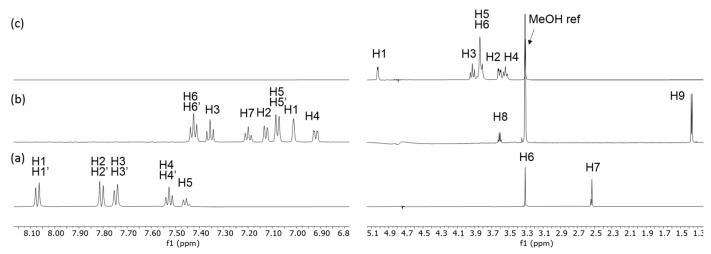
The ^1^H NMR assignation of (**a**) fenbufen, (**b**) fenoprofen, and (**c**) BCyD.

**Figure 3 ijms-24-07544-f003:**
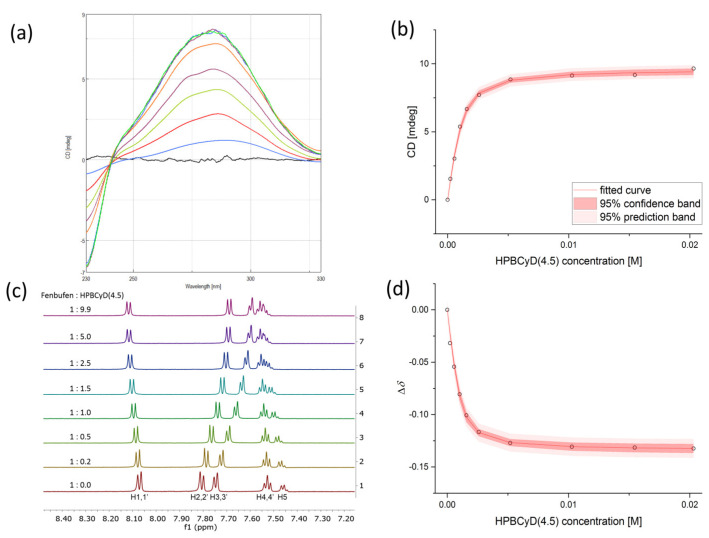
(**a**) Induced CD spectra of HPBCyD(4.5)-fenbufen complexes with increasing host concentration (highest host concentration corresponds to the tallest peak) in 90% H_2_O/10% D_2_O. (**b**) The regression model of the ellipticity data obtained from the CD measurements at *λ*_max_. (**c**) ^1^H NMR spectra series (aromatic region) of HPBCyD(4.5)-fenbufen complexes at different host–guest ratios in 90% H_2_O/10% D_2_O. (**d**) The regression model of the H2 NMR signal (obtained from global fit).

**Figure 4 ijms-24-07544-f004:**
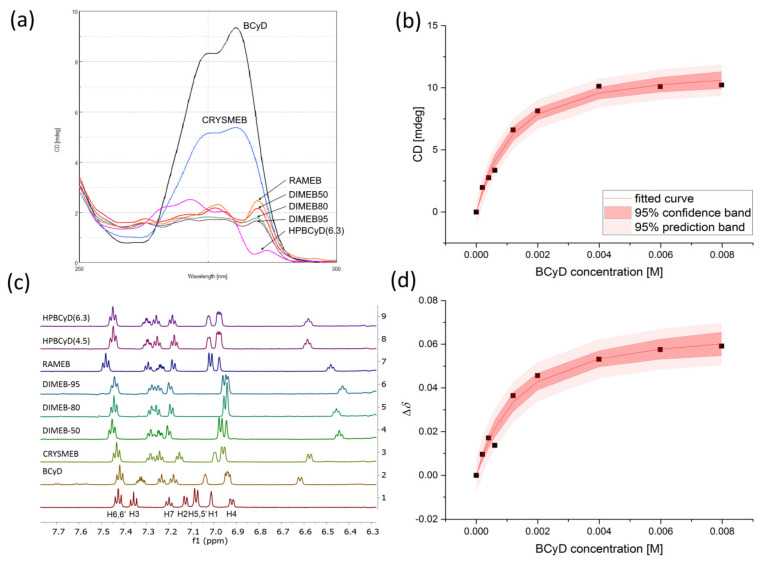
(**a**) The induced CD spectra of CyD-fenoprofen 1:10 in 90% H_2_O/10% methanol-*d*_4_. (**b**) The regression model of the ellipticity data for BCyD-fenoprofen obtained from the CD measurements at *λ*_max_. (**c**) The ^1^H NMR spectra of CyD-fenoprofen 1:10 complexes in 90% H_2_O/10% methanol-*d*_4_. (**d**) The regression model of the H2 NMR signal of BCyD-fenoprofen (obtained from global fit).

**Figure 5 ijms-24-07544-f005:**
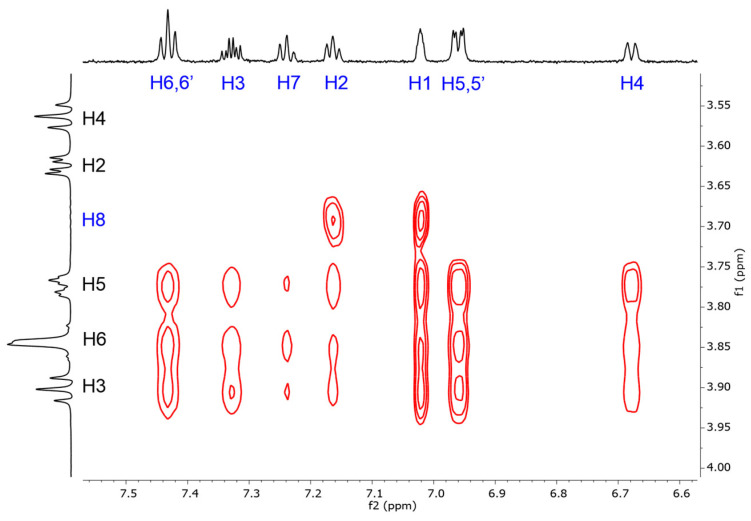
The ROESY NMR spectrum of fenoprofen-BCyD 1:5 complex in 90% H_2_O/10% methanol-*d*_4_. Assignation of the signals in blue is for fenoprofen, in black is for BCyD.

**Figure 6 ijms-24-07544-f006:**
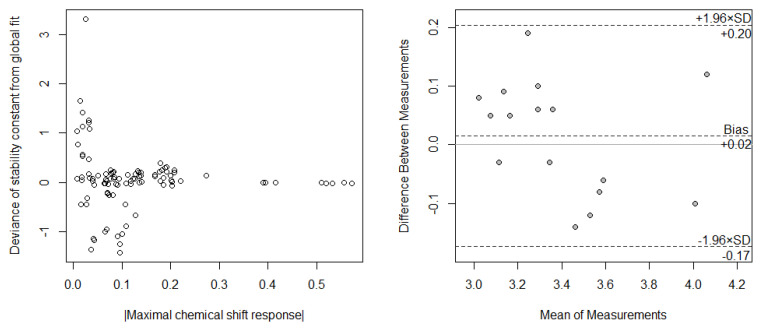
Left: The deviance of each log*K* obtained from single NMR signals from its concomitant globally fit log*K* as a function of absolute maximal chemical shift response. Right: The Bland–Altman plot of the complex stability constants for CyD-fenbufen and CyD-fenoprofen complexes obtained with two methods: CD and ^1^H NMR spectroscopy.

**Figure 7 ijms-24-07544-f007:**
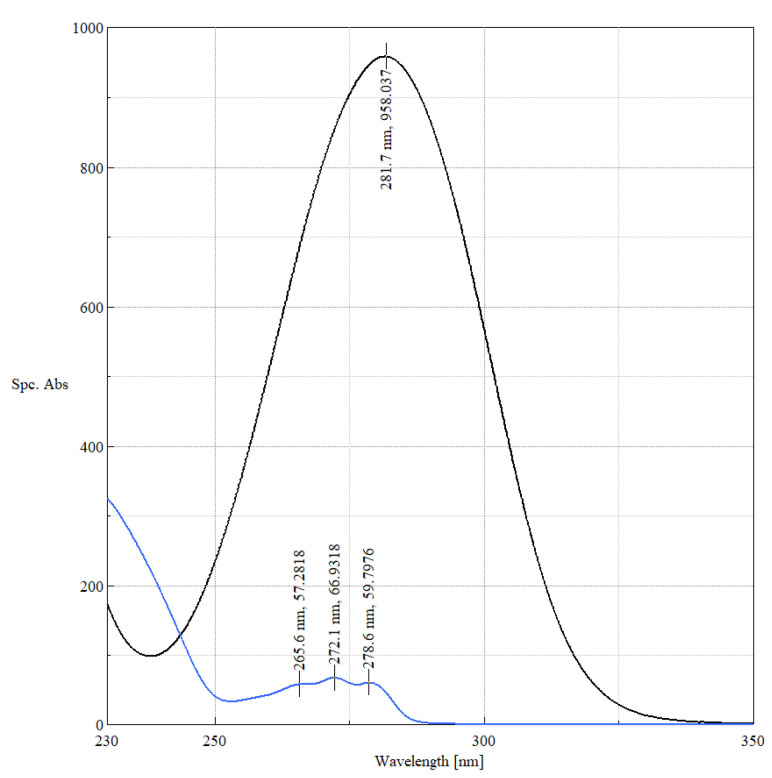
Specific absorptions of fenbufen (black) and fenoprofen calcium (blue) in methanolic solutions with extrema.

**Table 1 ijms-24-07544-t001:** Comparison of stability constants. Abbreviations: CE—capillary electrophoresis; PS—phase solubility measurement, NMR—^1^H nuclear magnetic resonance spectroscopy.

CyD/Guest Compound	log*K* Determined by CD [14]	log*K* from Literature
BCyD/		
Bifonazole	3.40	3.40–3.52 (CE) [23]
Clotrimazole	2.65	2.66 (PS) [24]
DIMEB/		
Tioconazole	3.86	3.84 (CE) [25]
HPBCyD/		
Bifonazole	4.46	3.66 (NMR) [26]
Clotrimazole	2.09	2.65 (PS) [27]
Miconazole	2.71	2.56 (PS) [28], 2.41 (PS) [29]
Tioconazole	3.30	2.86 (CE) [25]
SBEBCyD/		
Bifonazole	4.72	3.94 (NMR) [26]

**Table 2 ijms-24-07544-t002:** The complex stability constants (in log10 units ± standard error) of fenbufen with the various cyclodextrins determined by CD and NMR spectroscopy in 90% H_2_O/10% D_2_O.

Cyclodextrin	log*K*_CD_	log*K*_NMR_	Diff.
BCyD (H_2_O) ^1^	3.35 ± 0.03	NA	NA
BCyD	3.34 ± 0.02	3.24 ± 0.06	0.10
CRYSMEB	3.39 ± 0.08	3.33 ± 0.05	0.06
DIMEB50	3.56 ± 0.12	3.62 ± 0.07	−0.06
DIMEB80	4.12 ± 0.28	4.00 ± 0.14	0.12
DIMEB95	4.06 ± 0.07	3.95 ± 0.10	0.10
RAMEB	3.59 ± 0.08	3.47 ± 0.02	0.12
HPBCyD(4.5)	3.39 ± 0.04	3.53 ± 0.06	−0.14
HPBCyD(6.3)	3.53 ± 0.04	3.61 ± 0.06	−0.08

^1^ Initial trial experiment performed in H_2_O without any deuterated solvent.

**Table 3 ijms-24-07544-t003:** The complex stability constants of CyD-fenbufen complexes (every odd line) followed by the maximal chemical shift response (every even line) obtained from regression fits performed on the data of each individual NMR peak for CyD-fenbufen complexes. The uncertainty of chemical shift response values (on ppm scale) is on the order of 0.001.

Cyclodextrin	H1,1′	H2,2′	H3,3′	H4,4′	H5
BCyD	NA	3.41	3.30	2.95	2.70
		−0.179	−0.273	−0.128	−0.095
CRYSMEB	NA	3.31	3.37	3.19	3.23
		−0.185	−0.185	0.029	0.070
DIMEB50	NA	3.83	3.94	4.28	3.42
		−0.201	−0.189	0.034	0.109
DIMEB80	NA	4.09	4.06	3.81	3.55
		−0.207	−0.200	0.026	0.100
DIMEB95	NA	4.08	4.17	4.78	3.58
		−0.220	−0.208	0.014	0.090
RAMEB	NA	3.68	3.70	3.09	2.97
		−0.176	−0.183	0.041	0.095
HPBCyD(4.5)	2.93	3.62	3.59	NA	3.09
	0.036	−0.134	−0.168		0.065
HPBCyD(6.3)	3.10	3.71	3.67	NA	3.19
	0.044	−0.132	−0.167		0.069

**Table 4 ijms-24-07544-t004:** The complex stability constants (in log10 units ± standard error) of fenoprofen with the various cyclodextrins determined by CD and ^1^H NMR spectroscopy in 90% H_2_O/10% methanol-*d*_4_.

Cyclodextrin	log*K*_CD_	log*K*_NMR_	Diff.
BCyD (H_2_O) ^1^	3.12 ± 0.02	NA	NA
BCyD	3.06 ± 0.05	2.98 ± 0.04	0.08
CRYSMEB	3.10 ± 0.02	3.05 ± 0.01	0.05
DIMEB50	3.10 ± 0.08	3.13 ± 0.02	−0.03
DIMEB80	3.33 ± 0.09	3.36 ± 0.01	−0.03
DIMEB95	3.32 ± 0.09	3.26 ± 0.01	0.06
RAMEB	3.34 ± 0.07	3.15 ± 0.004	0.19
HPBCyD(4.5)	3.18 ± 0.08	3.09 ± 0.01	0.09
HPBCyD(6.3)	3.19 ± 0.05	3.14 ± 0.004	0.05

^1^ Initial trial experiment performed in H_2_O without any deuterated solvent.

**Table 5 ijms-24-07544-t005:** The complex stability constants of CyD-fenoprofen complexes (every odd line) followed by the maximal chemical shift response (every even line) obtained from regression fits performed on the data of each individual NMR peak for CyD-fenoprofen complexes. The uncertainty of chemical shift response values (on ppm scale) is on the order of 0.001.

Cyclodextrin	H1	H2	H3	H4	H5,5′	H6,6′	H7
BCyD	3.17	2.99	2.99	2.98	2.99	3.02	2.96
	0.026	0.068	−0.040	−0.390	−0.179	−0.008	0.043
CRYSMEB	3.10	3.09	3.02	3.05	3.05	3.38	3.11
	−0.018	0.033	−0.090	−0.415	−0.140	0.010	0.051
DIMEB50	3.16	3.11	3.16	3.12	3.16	3.20	3.16
	−0.076	0.088	−0.126	−0.556	−0.123	0.032	0.094
DIMEB80	3.46	3.25	3.43	3.35	3.41	3.60	3.45
	−0.076	0.071	−0.112	−0.519	−0.139	0.018	0.082
DIMEB95	3.39	3.19	3.31	3.29	3.39	3.53	3.40
	−0.079	0.074	−0.118	−0.532	−0.138	0.019	0.083
RAMEB	3.18	3.13	3.14	3.15	3.13	3.22	3.13
	−0.040	0.064	−0.136	−0.510	−0.072	0.068	0.109
HPBCyD(4.5)	NA	3.07	3.12	3.09	3.11	3.00	3.04
	NA	0.059	−0.071	−0.395	−0.118	0.027	0.063
HPBCyD(6.3)	2.94	3.15	3.13	3.14	3.12	3.34	3.16
	0.016	0.069	−0.066	−0.395	−0.118	0.031	0.067

## Data Availability

The data presented in this study are available in the article and Appendix A. The recorded spectra presented in this study are available on request from the corresponding author.

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
