# Peer review of "Effect of Substitution Degree and Homogeneity on Cyclodextrin-Ligand Complex Stability: Comparison of Fenbufen and Fenoprofen Using CD and NMR Spectroscopy"

_ijms, 2023, doi:10.3390/ijms24087544_

Round 1
Reviewer 1 Report
Manuscript: “Effect of substitution degree and homogeneity on cyclodextrin-2 ligand complex stability: comparison of fenbufen and 3 fenoprofen using CD and NMR Spectroscopy”
The authors present a comparison between ICD and NMR as methodologies for the evaluation of the interaction of two NSAID Drugs, fenbufen and fenoprofen, with cyclodextrins. In particular, they provide estimates of the stability constants of drug-cyclodextrin complexes and describe the effects of the degree of substitution and homogeneity/heterogeneity of substituents in the magnitude of stability constants.
Major comments:
i) The reasoning behind the choice of the two drugs employed in the study is not satisfactorily delineated. The work would benefit from a more intuitive discussion of this matter.
ii) Noticeably, the authors aim at demonstrating the potentiality of ICD as an alternative methodology to NMR for the evaluation of host-guest interactions. They provide evidence of ICD potential in the measurement of stability constants (Table 4) but should address in more detail the pitfalls of such methodology. They should discuss more thoroughly the advantages/disadvantages of both methodologies and refer some fundamental limitations of ICD methodology.
iii) The authors should provide more information concerning other parameters, besides measurement of stability constants, that can be evaluated by ICD. They provide ROESY data for demonstrating the formation of true inclusion complexes. Can ICD provide clear evidence of such formation or is the information less robust in this evaluation?
iv) The authors refer that “1H NMR spectra were referenced to the methanol (fenbufen solutions) or the methanol-d4 (fenoprofen solutions) signals”. Being the 1H NMR chemical shifts one of the most relevant measurements for the analysis performed in this work, the authors should have considered another referencing alternative (e.g., an insert with another compound that would not be affected by changes in solution properties due to changes in guest:CyD ratios).
Minor comments:
i) Page 9 line 269: “(NSAID:CyD ratio 1:0)”. A very strange form of referring to NSAID solutions deprived of CyD.
ii) Page 9 line 276: “For fenbufen the ellipticity data were read at the peak maximum, for fenoprofen at the spectral extrema due to the more structured spectra as appeared”. Needs editing.
iii) Page 10 line 305: “The regression analyses the non-linear curve fitting was performed”; Needs correction.
Author Response
Major comments:
- i) The reasoning behind the choice of the two drugs employed in the study is not satisfactorily delineated. The work would benefit from a more intuitive discussion of this matter.
As requested, we have added a more detailed discussion of choosing fenbufen and fenoprofen as model compounds to the relevant part of the introduction (p.2 line 77-85)
- ii) Noticeably, the authors aim at demonstrating the potentiality of ICD as an alternative methodology to NMR for the evaluation of host-guest interactions. They provide evidence of ICD potential in the measurement of stability constants (Table 4) but should address in more detail the pitfalls of such methodology. They should discuss more thoroughly the advantages/disadvantages of both methodologies and refer some fundamental limitations of ICD methodology.
On pages 4 and 5 following the reviewer's comment, the text has been completed between lines 114-117 and 136-148.
iii) The authors should provide more information concerning other parameters, besides measurement of stability constants, that can be evaluated by ICD. They provide ROESY data for demonstrating the formation of true inclusion complexes. Can ICD provide clear evidence of such formation or is the information less robust in this evaluation?
While NMR spectroscopy provides atomic-level structure information, ICD spectra are less informative, but the law developed by Kodaka and Harata ([30];[32]), the work of Kajtár [31] and the work of Allenmark (CHIRALITY 15:409–422 (2003) can be used to estimate the complex structure from spectra. This is now discussed in more details in lines 124-128.
- iv) The authors refer that “1H NMR spectra were referenced to the methanol (fenbufen solutions) or the methanol-d4 (fenoprofen solutions) signals”. Being the 1H NMR chemical shifts one of the most relevant measurements for the analysis performed in this work, the authors should have considered another referencing alternative (e.g., an insert with another compound that would not be affected by changes in solution properties due to changes in guest:CyD ratios).
Using methanol as a chemical shift reference standard in cyclodextrin complexation measurements by NMR is quite accepted because it does not have significant interaction with CyDs. To make it clear for the reader, we inserted some literature references mentioning usage of methanol reference in similar measurements in the Materials and Methods section. (Ref. 41-42) (lines 317-318)
Minor comments:
- i) Page 9 line 269: “(NSAID:CyD ratio 1:0)”. A very strange form of referring to NSAID solutions deprived of CyD.
The sentence has been corrected. (line 321)
For all cases, the spectrum of an aqueous solution containing the same concentration of NSAID without CyD was also recorded.
- ii) Page 9 line 276: “For fenbufen the ellipticity data were read at the peak maximum, for fenoprofen at the spectral extrema due to the more structured spectra as appeared”. Needs editing.
The sentence has been corrected. (line 331)
For fenbufen the ellipticity data were read at the peak maximum, for fenoprofen at the 2 or 3 peak maxima (the number of maxima depends on the type of the CyD).
iii) Page 10 line 305: “The regression analyses the non-linear curve fitting was performed”; Needs correction.
The sentence has been corrected. (line 365)
The non-linear regression analyses were performed using Origin Pro 8
Reviewer 2 Report
This paper presents the measurements of the stability constants of cyclodextrin derivatives by the CD and NMR spectroscopy. The methodology looks fine, with sound results well written up. I have several comments for improvement of the manuscript.
1. Why fenbufen and fenoprofen? The authors may remark on the motivation of choosing these two guest compounds.
2. Although the structures of the studied cyclodextrin derivatives may be more difficult to determine than the measurements of stability constants, they still are of much interest to some readers. The authors briefly commented on the structures at the end of the manuscript, but it seems to be desirable to elaborate more on this subject, along with the related Figures and References.
3. Comparison with the stability constants of other systems in literature would be helpful. I recommend the authors to do it with some plausible discussions on the differences and trends of the stability constants.
4. Please cite the following related papers on cyclodextrins and derivatives:
General:
Crini, G.; Review: A History of Cyclodextrins, Chem. Rev. 2014, 114, 10940-10975.
R. C. Dunbar, J. D. Steill, J. Oomens, J. Am. Chem. Soc., 2011, 133, 1212–1215.
J. Szejtli, Chem. Rev., 1998, 98, 1743–1754.
Structural determination:
H. Choi, Y.‑H. Oh, S. Park, S.‑S. Lee, H. B. Oh, S. Lee, Sci. Rep. 2022, 12, 8169.
C. B. Lebrilla, Acc. Chem. Res. 2001, 34, 653–661.
Z. Li, E. P. A. Couzijn and X. Zhang, J. Phys. Chem. B, 2012, 116, 943–950.
Chiral differentiation:
Lee, S.-S.; Lee, J.-u.; Oh, J. H.; Park, S.; Hong, Y.; Min, B. K.; Lee, H. H. L.; Kim, H. I.; Kong, X.; Lee, S.; Oh, H. B. Phys. Chem. Chem. Phys. 2018, 20, 30428- 30436.
A. Filippi, C. Fraschetti, S. Piccirillo, F. Rondino, B. Botta, I. D’Acquarica, A. Calcaterra and M. Speranza, Chem. Eur. J. 2012, 18, 8320–8328.
A. Sen, K. L. Barbu-Debus, D. Scuderi and A. Zehnacker-Rentien, Chirality, 2013, 25, 436–443.

Author Response
- Why fenbufen and fenoprofen? The authors may remark on the motivation of choosing these two guest compounds.
As requested, we have added a more detailed discussion of choosing fenbufen and fenoprofen as model compounds to the relevant part of the introduction (p.2 line 76-83)
- Although the structures of the studied cyclodextrin derivatives may be more difficult to determine than the measurements of stability constants, they still are of much interest to some readers. The authors briefly commented on the structures at the end of the manuscript, but it seems to be desirable to elaborate more on this subject, along with the related Figures and References.
Text has been added at lines 201-227
- Comparison with the stability constants of other systems in literature would be helpful. I recommend the authors to do it with some plausible discussions on the differences and trends of the stability constants.
We have supplemented the Conclusion section with the requested discussion. (Lines 385-389)
- Please cite the following related papers on cyclodextrins and derivatives:
Some of the suggested references have been added to the manuscript, [5; 10, 12, 13 ] apart from the publication of Szejtli, as it already appears in the manuscript as the first reference.
Dunbar’s, Filippi’s and Sen’s publications are not relevant to the present work therefore we have added new relevant references. [6; 11; 14; 15]
Reviewer 3 Report
The present paper described an effect of substitution degree and homology on CD-ligand complex stability using CD and NMR spectroscopy.
The following points should be considered for publication:
(1) In Fig. 2, 3, 4 and Tables 2, 4, the sample solution was 90%H2O and 10% D2O. The general solution used for NMR is 100% D2O. The reason used the aforementioned solution should be explained.
(2) A structure of BCyD with 1H numbering should be shown to interpret the ROESY spectrum shown in Fig. 4.
(3) The 1H NMR spectra of fenbufen, fenoprofen and BCyD together with assignments should be shown.
(4) In Fig. 4, 1H signal resonating at 6.4-6.5 ppm should be assigned.
(5) The final concentration of sample solutions used in NMR and CD measurements should be clearly described in Materials and Methods section.
(6) The detailed condition of NMR and CD, such as transient in ROESY experiment, should be described.
(7) The purpose of the current research is not clear. Did the authors compare with the past results performed by other group? The conclusion section should be corresponding to the purpose of the research.
Author Response
(1) In Fig. 2, 3, 4 and Tables 2, 4, the sample solution was 90%H2O and 10% D2O. The general solution used for NMR is 100% D2O. The reason used the aforementioned solution should be explained.
As we discussed in the Materials and Methods sections we wanted to use exactly the same solutions for both the CD and NMR experiments. In CD experiments the commonly used solvent is 100% H2O, but since in NMR experiments stabilization of the frequency requires deuterium lock signal we had to use at least a small amount (10%) deuterated solvent in the solutions. We could have used 100% deuterated solvents in the CD experiments as well, but regarding that we needed to prepare several solutions, using only deuterated solvents seemed to be a waste of deuterated solvents. Nevertheless, we inserted an explanation of using 10% deuterated solvents in the text. (line 316)
(2) A structure of BCyD with 1H numbering should be shown to interpret the ROESY spectrum shown in Fig. 4.
Figure 1 was updated to show the structure of BCyD as well.
(3) The 1H NMR spectra of fenbufen, fenoprofen and BCyD together with assignments should be shown.
A new figure (Figure 2) was inserted to show the NMR assignments.
(4) In Fig. 4, 1H signal resonating at 6.4-6.5 ppm should be assigned.
The blue signal which could be seen in Fig. 4. at 6.61/3.57 ppm is an in-phase (anti-phase to the ROESY cross peaks) noise signal therefore we have performed additional phase- and baseline corrections on the 2D ROESY spectrum to get rid of the disturbing noise signal in this valuable region of the spectrum and replaced the original Figure with the new one. Furthermore, we have inserted the missing assignation of two BCyD signal (H2 and H4 between 3.55-3.65 ppm) on the 1D spectrum of the f1 axis.
(5) The final concentration of sample solutions used in NMR and CD measurements should be clearly described in Materials and Methods section.
Since the concentrations of the final solutions were a bit different in every solution series the exact concentrations are not listed in the Materials and Method section but both the precise CyDs’ and guests’ concentrations can be found in the Supplementary Tables S1-S23. To call the reader’s attention to this information, we have inserted a reference to the Supplementary Tables in the Materials and Methods section. (lines 322-323)
(6) The detailed condition of NMR and CD, such as transient in ROESY experiment, should be described.
We have supplemented the Materials and Methods section with the requested information. (lines 336-337; 343-345)
(7) The purpose of the current research is not clear. Did the authors compare with the past results performed by other group? The conclusion section should be corresponding to the purpose of the research.
To make the purpose of the research clearer for the readers, the last paragraph of the introduction section was expanded (Lines:141-148) and the conclusion section was elaborated accordingly. (lines 354-355; 393-399)
Round 2
Reviewer 1 Report
The authors made significant efforts towards answering raised concerns. Its fine to be accepted in its current form.
Reviewer 2 Report
I am satisfied with the revised manuscript. Now I think it is publishable.
Reviewer 3 Report
The revised manuscript would meet the criteria of the journal.